# The Effect of Citicoline on the Expression of Matrix Metalloproteinase-2 (MMP-2), Transforming Growth Factor-β1 (TGF-β1), and Ki-67, and on the Thickness of Scleral Tissue of Rat Myopia Model

**DOI:** 10.3390/biomedicines10102600

**Published:** 2022-10-17

**Authors:** Eka Wahyuningsih, Dimas Wigid, Astrid Dewi, Hariwati Moehariadi, Hidayat Sujuti, Nanda Anandita

**Affiliations:** Department of Ophthalmology, Faculty of Medicine, Universitas Brawijaya, Dr. Saiful Anwar General Hospital, Malang 65111, Indonesia

**Keywords:** axial elongation, citicoline, MMP-2, TGF-β1, scleral thickness, fibroblast proliferation, rat myopia model

## Abstract

Citicoline, presumed to be involved in the dopaminergic pathway, might play a role as a candidate agent in controlling myopia. However, its study with respect to myopia is limited. The aim of this study is to demonstrate the effect of citicoline on the expression of MMP-2, TGF-β1, and Ki-67, and on the thickness of scleral tissue of a rat myopia model. Immunohistochemistry was performed to evaluate the expression of MMP-2, TGF-β1, and Ki-67 as the markers for fibroblast proliferation. Hematoxylin and eosin staining were used to evaluate scleral thickness. An electronic digital caliper was used to evaluate the axial length. The treatment group administered with 200 mg/kg BW/day had the lowest mean MMP-2 expression, axial elongation, and fibroblast proliferation, but it had the highest mean scleral thickness. The treatment group administered with 300 mg/kg BW/day had the highest mean TGF-β1 expression. Citicoline is able to decrease MMP-2 expression and fibroblast proliferation and increase TGF-β1 expression and scleral tissue thickness significantly in the scleral tissue of rat models for myopia.

## 1. Introduction

Myopia is the most important cause of visual impairment throughout the world, affecting about one half of the world’s population. Although it is possible to correct myopic refractive errors, myopia’s progression is often accompanied by serious ocular disorders such as retinal detachment, macular hemorrhage, macular degeneration, and glaucoma. These complications can cause irreversible vision loss and even blindness. Therefore, there is a very urgent need to find the most effective treatment for controlling myopia’s progression, while minimizing side effects [1,2].

Management of myopia currently includes optics, pharmacological interventions, and refractive surgery. One of the pharmacological therapy choices to slow progression is topical therapy with muscarinic acetylcholine receptor antagonists, such as atropine, which has reduced progression by about 90%, and pirenzepine up to 44%. However, both agents have side effects, which are blurred vision at close range, photophobia, glare, and recurrent allergic blepharitis [3,4]

Apomorphine, levodopa, and citicoline, which are all involved in the dopaminergic pathway, may play a role as candidate agents in controlling myopia’s progression. Citicoline has beneficial effects in the treatment of glaucoma and amblyopia, but studies on its effects on myopia are limited. Citicoline (CDP-choline) is able to activate the biosynthetic pathway of structural phospholipids in neuronal membranes and stimulate several neurotransmitter systems, including the dopaminergic system. After the neurotransmitter is released, it diffuses laterally to the retina and choroid [5,6,7].

A study by Mao reported that there was a significant increase in retinal dopamine levels in guinea pigs visually deprived for 10 days and injected with intraperitoneal citicoline (500 mg/kg BW, twice a day). Their results demonstrated that an intraperitoneal injection of citicoline could retard the myopic shift induced by form deprivation in guinea pigs, which was mediated by an increase in the retinal dopamine levels [6]. Debora et al. reported that the administration of a dopamine agonist, a chemical mediator of visual stimuli, leads to choroidal thickening and the inhibition of axial elongation [8].

The most common structural abnormality associated with myopia is an elongation of the posterior segment of the eyeball [6]. Axial elongation is closely related to the biomechanical properties of the sclera, which, in turn, are largely dependent on transforming growth factor-β1 (TGF-β1) regulation and extracellular matrix metabolism [5,6,7]. Scleral remodeling can lead to exaggerated eye growth, causing myopia, for which MMP-2 has a major role in this process. The sclera undergoes remodeling, namely, changes in fibroblast synthesis and degradation of the extracellular matrix [6,9]. Ki-67 is a sensitive, predictive marker that operates as a biomarker for proliferation, including that of fibroblasts. This marker is also used to detect and be one of the prognostic factors in tumor cases. This protein is excreted at all phases of the cell cycle except in G0 [9,10].

It is important to study effective therapies to inhibit the progression of myopia while minimizing the side effects. This study aims to investigate the effect of citicoline on the expression of MMP-2, TGF-β1, and Ki-67 as fibroblast cell-proliferating markers, and on the thickness of scleral tissue of rat models for myopia. We propose that citicoline reduces MMP-2 expression and fibroblast proliferation and increases TGF-β1 expression and scleral tissue thickness and, therefore, may be useful for inhibiting the progression of myopia.

## 2. Materials and Methods

### 2.1. Research Design

This study has received ethical approval from the Animal Care and Use Committee of Universitas Brawijaya, Indonesia, with ethical clearance no. 1006-KEP-UB. The first study was conducted from October 2018 to January 2019. We measured the axial length and the expression of MMP-2 in the first study. The second study was conducted from January to July 2020. We observed the axial length and the expression of TGF-β1 and Ki-67 in the second study. Both of the studies were conducted in the Bioscience Laboratory of Universitas Brawijaya, Indonesia.

### 2.2. Population and Sample

The design of this study was an in vivo experiment of rat sclera administered with citicoline. The inclusion criteria involved four-week-old male Wistar rats weighing 150–200 ± 15 g, who were healthy, characterized by active movements, and possessed clear eyes and shiny thick white fur. Wistar rats were obtained and maintained at the Bioscience Laboratory of Universitas Brawijaya.

In the first study, the male Wistar rats (n = 25) were divided into five groups, namely, (a) negative control group (n = 5)—no lens or other treatment; (b) positive control group (n = 5)—an S -10.00 D lens attached but not treated with citicoline; (c) treatment group 1 (n = 5)—an S -10.00 D lens + 50 mg/kg BW/day of citicoline; (d) treatment group 2 (n = 5)—an S -10.00 D lens + 100 mg/kg BW/day of citicoline; and (e) treatment group 3 (n = 5)—an S -10.00 D lens + 200 mg/kg BW/day of citicoline.

In the second study, the male Wistar rats (n = 25) were divided into five groups, namely, (a) negative control group (n = 5)—no lens or other treatment; (b) positive control group (n = 5)—an S -10.00 D lens attached but not treated with citicoline; (c) treatment group 1 (n = 5)—an S -10.00 D lens + 100 mg/kg BW/day of citicoline; (d) treatment group 2 (n = 5)—an S -10.00 D lens + 200 mg/kg BW/day of citicoline; and (e) treatment group 3 (n = 5)—an S -10.00 D lens + 300 mg/kg BW/day of citicoline.

Rats that died during the course of this study, caused by improper oral gavage treatment, or experienced infection or ocular inflammation caused by periorbital suturing, were excluded.

### 2.3. Acclimatization

Acclimatization of experimental animals was carried out in cages in the Pharmacology Laboratory of Brawijaya University Malang for 5 days. The experimental animals were kept in cages measuring 20 cm × 30 cm × 40 cm, where each cage contained 1 rat in a room with a temperature of 22–25 °C and a light–dark cycle of 12/12 h. The experimental animals’ food was in the form of pellets, and they were given drinking water from the PDAM (the state-owned water utility company). During the implementation of this study, the rats were treated carefully, and the ethical feasibility of the study was adhered to regarding experimental animals.

### 2.4. Induction of Myopia and Administration of Citicoline

Lens-induced myopia (LIM) modelling was carried out on Wistar rats with an S -10.00 D lens attached on the periorbital area of each rats’ right eye. To induce myopia using lens-induced myopia (LIM) method, the rats were anesthetized with 0.03 mL of ketamine and 0.02 mL of intramuscular xylazine. The hair in the periorbital area of the right eye was shaved prior to disinfection. After disinfection, the CR 39 multicoat lens, with a power of S -10.00 D and 10 mm in diameter with a 14 mm rubber frame, was sutured with 3-0 mersilk on the right periorbital area (Figure 1a). Chloramphenicol ointment was applied to the wound after suturing. 

This study used a citicoline (RG Choline, PT. Kalbe Farma Tbk, Indonesia) dose of 1000 mg for each tablet. After two weeks of myopic induction, each treatment group was administered citicoline at a dosage of 50, 100, 200, or 300 mg/kg BW/day. The rats were individually weighed using an electronic balance to determine the amount of citicoline given in accordance with the dosage decided in each group. Citicoline was turned into powder and dissolved in sterile distilled water. Citicoline was given once a day by oral gavage for seven days; the doses were adjusted to the rat’s body weight.

### 2.5. Sample Collections and Measurement of Axial Length

After three weeks of study, all of the rats were put down. In the positive control and treatment groups, the suture fixing the lens was removed. The periorbital skin was dissected, and the right eye was enucleated. The axial lengths in the negative and positive control groups were measured. Axial length was measured for the largest anteroposterior diameter starting from the cornea to the posterior poles with electronic digital calipers (0.01 mm in accuracy, Krisbow) (Figure 1b). Measurements were performed by three observers, each of whom carried out measurements three times. Measurements were carried out in a blind manner by measuring from the anterior surface to the posterior surface of the eye. The results were presented in the form of a mean length with standard deviation. After the measurements were completed, all the rat eyeballs were preserved using formalin.

### 2.6. Measurement of The Thickness of Scleral Tissue

The slides were colored with Hematoxylin-Eosin (HE) staining using xylol, alcohol of various concentrations, water, Mayer’s HE, and Eosin 0.5%-alcohol-acetic acid. Then, the scleral thickness was measured from the scleral tissue, which is the transparent opaque tissue lining the outer part of the eye taken from the rats’ enucleated eyes that passed the HE-staining procedure. Scleral tissue thickness measurements were carried out four times under light microscopy using a 600× magnification in the anterior and posterior poles. Scleral thickness was measured by drawing a perpendicular line from the outermost layer to the innermost layer of the sclera. 

### 2.7. Measurement of MMP-2, TGF-β1, and Ki-67 Expression Using Immunohistochemistry

The slides were heated in an incubator (60 °C for 60 min) and then washed in various concentrations of ethanol and rinsed in sterile distilled water before the deparaffinization process. The slides were placed in 0.01 M sodium citrate buffer at pH 6.0 and microwaved at a high temperature until the buffer boiled. At room temperature, 3% peroxidase in methanol, primary antibody (MMP-2, TGF-β1, and Ki-67), biotin-labeled secondary antibody (biotin-labeled anti-rabbit IgG), and 3,3-Diaminobenzidine Tetratetrahydrochloride (DAB chromagen: DAB buffer = 1:50) were sequentially added to the slides with specific incubation times and rinsed with distilled water in each addition. Counterstain with hematoxylin was applied to the slides. The slides were incubated at room temperature for 5–10 min, rinsed with tap water, and dried. After the addition of mounting media, the slides were closed with a coverslip.

The slides were observed using an Olympus BX53 light microscope. Then, this study measured MMP-2, TFG-β1, and Ki-67 expressions in the scleral tissues, which were the transparent opaque tissues lining the outer part of the eye taken from the enucleated rats’ eyes that passed the paraffinization process. Subsequently, immunohistochemical staining was carried out. The MMP-2, TGF-β1, and Ki-67 expressions are displayed by the brown staining of target protein in rat scleral tissue’s cell nucleus and cytoplasm, and they were assessed quantitatively using the ImmunoRatio software and presented in percentages. Observations under 600× magnification were carried out on four visual fields. The MMP-2 and TGF-β1 expressions were observed in the posterior poles on both sides of the optic nerve. The Ki-67 expression was examined in the anterior sclera and posterior poles.

### 2.8. Statistical Analysis

SPSS 22.0 software was used to analyze the data. All data were expressed as mean ± SD and analyzed by independent *t*-test, one-way ANOVA, Tukey’s posthoc hypothesis test, Pearson correlation, and simple linear regression. A *p*-value less than 0.05 is considered statistically significant. 

## 3. Results

### 3.1. Axial Length and Histological Structure of Scleral Tissue

After 3 weeks, the measurements of the axial length of the eyeballs had been performed in all groups. Comparisons were made with regard to the axial lengths of the negative control group. Figure 2a shows the axial lengths for the five groups in the first study. Based on the one-way ANOVA test, it was found that the *p*-value for axial elongation was *p* = 0.000, and based on the Tukey test, there were several treatment groups that showed differences from the positive control at the treatment dosage of 100 mg/kg BW (5.994 ± 0.023 vs. 5.836 ± 0.113, *p* = 0.032) and 200 mg/kg BW (5.994 ± 0.023 vs. 5.742 ± 0.066, *p* = 0.000). Whereas, at a dose of 50 mg/kg BW, no significant difference was found (5.994 ± 0.023 vs. 5.884 ± 0.055, *p* = 0.208). 

Figure 2b shows the axial lengths for the five groups in the second study. The negative control group (n = 5) shows a shorter axial length than that of the positive control group (n = 5) using the LIM method (5.45 ± 0.07 vs. 6.04 ± 0.08, *p* = 0.000), the treatment group administered with 100 mg/kg BW/day (5.45 ± 0.07 vs. 5.98 ± 0.05, *p* = 0.000), the treatment group administered with 200 mg/kg BW/day (5.45 ± 0.07 vs. 5.64 ± 0.02, *p* = 0.000), and the treatment group administered with 300 mg/kg BW/day (5.45 ± 0.07 vs. 5.57 ± 0.03, *p* = 0.008). There is a significant difference in the axial length between the positive control group and the treatment group administered with 200 mg/kg BW/day (6.04 ± 0.08 vs. 5.64 ± 0.02, *p* = 0.000) and the treatment group administered with 300 mg/kg BW/day (6.04 ± 0.08 vs. 5.57 ± 0.03, *p* = 0.000). 

### 3.2. MMP-2 Expression

The immunohistochemical staining for MMP-2 expression showed that the rat models for myopia without treatment (positive control, n = 5) display a high number of MMP-2-positive stains (Figure 3B). In contrast, the control rats (negative control, n = 5) and rat models for myopia with a citicoline dosage of 100 mg/kg BW/day (n = 5) and 200 mg/kg BW/day (n = 5) lack evidence of MMP-2-positive staining (Figure 3A,D,E). These results were confirmed by a quantitative analysis of the MMP-2-positive area. 

In the negative control group, the degree of MMP-2 expression is lower than in the positive control group (15.87 ± 6.17 vs. 70.88 ± 18.28, *p* = 0.000). Among the treatment groups, the MMP-2 expressions are lower in the treatment groups with 100 mg/kg BW/day (31.64 ± 23.28 vs. 70.88 ± 18.28, *p* = 0.004) and the treatment groups with 200 mg/kg BW/day (18.38 ± 7.75 vs. 70.88 ± 18.28, *p* = 0.000) compared to the positive control group. 

### 3.3. Scleral Tissue Thickness

Measurements of scleral tissue thickness were carried out under light microscopy using a 600× magnification in the anterior and posterior scleral regions. In Figure 4 and Table 1, the thickness of the rats’ scleral tissues in the five groups can be observed. A significant difference in scleral thickness is shown between the negative control and the positive control groups (98.94 ± 6.06 μm vs. 80.74 ± 1.39 μm, *p* = 0.002 in the anterior sclera and 111.12 ± 2.99 μm vs. 59.84 ± 2.02 μm, *p* = 0.000 in the posterior pole). The treatment group administered with 100 mg/kg BW/day show a non-significant difference with respect to the positive control group (85.68 ± 4.17 μm vs. 80.74 ± 1.39 μm, *p* = 1.000 in the anterior sclera and 60.44 ± 6.18 μm vs. 59.84 ± 2.02 μm, *p* = 0.999 in the posterior pole). The treatment group administered with 200 mg/kg BW/day (93.10 ± 1.93 μm vs. 80.74 ± 1.39 μm, *p* = 0.002 in the anterior sclera and 85.12 ± 5.62 μm vs. 59.84 ± 2.02 μm, *p* = 0.000 in the posterior pole) and 300 mg/kg BW/day (90.95 ± 4.07 μm vs. 80.74 ± 1.39 μm, *p* = 0.000 in the anterior sclera and 84.24 ± 3.15 μm vs. 59.84 ± 2.02 μm, *p* = 0.000 in the posterior pole) show significant differences compared to the positive control group.

### 3.4. TGF-β1 Expression

Figure 5 describes the expression of TGF-β1 in the five groups using an immunohistochemical staining procedure that was analyzed using a light microscope. Based on the observations made with a light microscope, there is a difference in TGF-β1 expression among the groups. The immunohistochemical staining for TGF-β1 expression shows that the negative control group (n = 5) and the treatment group administered with 300 mg/BW/day display high levels of TGF-β1-positive stains in the anterior and posterior sclera (Figure 5A,E). In contrast, the rat models for myopia without treatment (positive control, n = 5) and the rat models for myopia with citicoline dosages of 100 mg/BW/day (n = 5) and 200 mg/kg BW/day (n = 5) lack TGF-β1-positive stains in the anterior and posterior sclera (Figure 5B–D). 

This study observed the TGF-β1 expression in the anterior and posterior sclera. In the anterior sclera, the negative control group shows a higher expression than the positive control group (4.55 ± 0.77 vs. 3.34 ± 0.54, *p* = 0.838) but there is no significant difference. Among the treatment groups, the treatment group administered with 300 mg/BW/day shows the highest level of TGF-β1 expression compared to the negative control group (12.86 ± 2.43 vs. 4.55 ± 0.77, *p* = 0.000), the positive control group (12.86 ± 2.43 vs. 3.34 ± 0.54, *p* = 0.000), the treatment group administered with 100 mg/kg BW/day (12.86 ± 2.43 vs. 2.26 ± 0.80, *p* = 0.000), and the treatment group administered with 200 mg/kg BW/day (12.86 ± 2.43 vs. 3.77 ± 3.15, *p* = 0.000).

In the posterior sclera, the negative control group shows a higher expression (5.01 ± 2.94, *p* > 0.05) than the positive control group (5.01 ± 2.94 vs. 3.40 ± 1.75, *p* = 0.905), but there is no significant difference. Among the treatment groups, the treatment group administered with 300 mg/BW/day shows the highest level of TGF-β1 expression compared to the negative control group (10.71 ± 4.56 vs. 5.01 ± 2.94, *p* = 0.042), the positive control group (10.71 ± 4.56 vs. 3.40 ± 1.75, *p* = 0.006), the treatment group administered with 100 mg/kg BW/day (10.71 ± 4.56 vs. 3.12 ± 1.20, *p* = 0.005), and the treatment group administered with 200 mg/kg BW/day (10.71 ± 4.56 vs. 3.30 ± 3.00, *p* = 0.006). 

### 3.5. Fibroblast Proliferation

An evaluation of fibroblast proliferation in the scleral tissue was conducted for all groups after the staining employing the immunohistochemical method using Ki-67: sc-23900 polyclonal antibody was performed. After the staining, we observed quantitatively the expression of fibroblast proliferation with a brownish density in the nucleus and cytoplasm of the fibroblast cells. In Figure 6 and Table 2, the levels of fibroblast proliferation of the scleral rat tissue can be observed in the five groups.

A non-significant difference in fibroblast proliferation is shown between the negative control group and the positive control group (3.81 ± 0.17 vs. 4.12 ± 0.18, *p* = 0.227) regarding the anterior sclera. Whereas there is a significant difference between the negative control group and the positive control group (3.53 ± 0.16 vs. 6.23 ± 0.14, *p* = 0.000) with respect to the posterior sclera. In the anterior sclera, the positive control group shows a higher level of fibroblast proliferation compared to the treatment group administered with 200 mg/kg BW/day (4.12 ± 0.18 vs. 3.46 ± 0.33, *p* = 0.001). In the posterior sclera, the positive control group shows a significant difference compared to the treatment groups administered with dosages of 100, 200, and 300 mg/kg BW/day (6.23 ± 0.14 vs. 5.43 ± 0.16 vs. 4.64 ± 0.44 vs. 4.96 ± 0.27, respectively, *p* < 0.005). Among the treatment groups, the treatment group administered with 200 mg/kg BW shows the lowest level of fibroblast proliferation.

## 4. Discussion

### 4.1. Elongation of the Eyeball of Rats after the Lens-Induced Myopia (LIM) Treatment

Myopia can be induced in rats using form-deprivation myopia (FDM) or lens-induced myopia (LIM) methods [11,12]. Visual deprivation during the early post-natal period reduces visual acuity in rats as in other vertebrates [13]. In the first study and the second study, it was found that the positive control group experienced an axial length elongation that was statistically significant compared to the negative control group. Axial length measurement is the secondary outcome of this study. 

Among the treatment groups in the initial study, there was no significant difference in axial length elongation after the administration of 50 mg/kg BW of citicoline, which could be caused by the insignificant dose range used. Remarkable differences in axial length elongation in the treatment groups with a citicoline dosage of 100 mg/kg BW and 200 mg/kg BW compared to the positive control group reveal the potential of citicoline in controlling the progression of myopia. The lower mean axial length elongation in the treatment group administered with 200 mg/kg BW compared to the negative control group may be influenced by biological variations among rats, interobserver bias, and the accuracy of the measuring instrument.

The research conducted by Jiang et al. (2018) reported that changes in the refractive disorders in the FDM varied from −2.00 D to −9.00 D, while changes in the elongation of the eyeball axis varied from 40 to 80 μm at 2–3 weeks. The varying elongation of the eyeball from studies regarding the induction of myopia can be affected by the different types of research methods employed in each study, including the use of the LIM or FDM method, the strength of the negative spherical lens diopters used, the duration of the study, the measurement instrument used, and the treatment of each study [12].

Various studies regarding ocular changes that occur after the induction of myopia have stated that the change in axial length that takes place can be an ocular response due to a blurred image captured by the retina. This situation happens continuously and will then be responded to by the retina and trigger the release of neurotransmitters by the retinal amacrine cells, which will convey a signal to the choroid and the sclera. In the progressive process of myopia, the structure of the sclera, as the outermost layer of the eye, is a key factor in initiating changes in the size of the eyeball. The main characteristic of the changes in the structure of the sclera of myopic individuals is a significant thinning of the sclera, especially in the posterior pole. These morphological changes are related to the presence of biochemical and mechanical changes in which the reduced collagen and proteoglycan content in the myopic eye cause the eyeball to stretch more easily, in turn causing an increase in the length of the eyeball’s axis [14,15,16]. 

### 4.2. The Effect of Citicoline on the Thickness of Rats’ Scleral Tissue 

The evaluations of the scleral tissue using a light microscope with a 600× magnification showed that there were differences among the groups in terms of the diameter and arrangement of collagen fibers. Mc Brien reported that there was evidence of scleral thinning and tissue loss, particularly in the posterior pole, in three shrews after 12 days of form deprivation. The changes were strongly associated with axial length elongation and the development of myopia. After 3–20 months of deprivation, a significant reduction in the distribution of the collagen fibril diameters was found, which is in line with findings in humans [17].

Based on the results of this study, in the negative control group, it was discovered that the thickness of the scleral tissue was thick both in the anterior and posterior sclera, with the thickness of the posterior pole thicker than the anterior sclera region. In the positive control group and the treatment group using various doses of citicoline, it was discovered that the scleral tissue in the anterior sclera and posterior pole appeared thinner than in the negative control group. However, the thicknesses of the scleral tissue in both the anterior scleral region and the posterior pole region in the treatment groups administered with 200 mg/kg BW/day of citicoline and those administered 300 mg/kg BW/day are thicker than in the positive control group. Using a light microscope, a difference in the thickness of the scleral tissue among the three treatment groups was observed. At the treatment dose of 100 mg/kg BW/day, the thickness of the scleral tissue almost resembled the positive control group. Whereas in the treatment group administered with citicoline at a dose of 200 mg/kg BW/day and 300 mg/kg BW/day, there was a significant increase in the thickness of the scleral tissue compared to the positive control group and the treatment group administered with citicoline at a dose of 100 mg/kg BW/day.

These results are in accordance with the study conducted by Morgan et al. (2013), who reported that there was scleral thinning and tissue loss after 12 days of deprivation. This condition mainly occurs in the posterior pole in experimental animals. These changes are related to the elongation of the eyeball axis and the development of myopia [11,18].

As for the Tukey’s test, in the treatment groups that received citicoline at a dose of 200 mg/kg/day and 300 mg/kg/day, there was a significant increase in the thickness of the scleral tissue compared to the positive control group, both in the anterior region and in the posterior pole. A citicoline administration affects dopamine concentrations by amplifying dopaminergic signaling pathways and the amount of dopamine released by neurons. The increase in dopamine synthesis may be due to the effect of citicoline through tyrosine hydroxylase activity. In in vivo studies, the activation of tyrosine hydroxylase led to the inhibition of dopamine reuptake at the synapse resulting in an increase in dopamine levels [19,20]. 

Dopamine is the main neurotransmitter in the mammalian retina, where it acts through synapses and diffuses to more distant targets. Ganglion cells also use dopamine to communicate with the central visual area. In a double-blind placebo-controlled study using electrophysiological methods (visually evoked potential and PERG), retinal and post-retinal functional improvements were observed in glaucoma patients treated with citicoline. The stimulation of the dopaminergic system by citicoline is a major mechanism in the improvement of the motor symptoms of Parkinson’s disease and the retinal and post-retinal function of the visual pathways of glaucoma patients. The evidence for this hypothesis is a significant increase in the retinal dopamine concentrations in rabbits treated with parenteral citicoline. The administration of CDP-choline as a dopamine receptor agonist is expected to decrease fibroblast growth factor and inhibit MMP-2 activity such that there is a decrease in the number of scleral fibroblasts and a reduction in the degradation of the extracellular matrix. From these conditions, it is expected that there will be an improvement in the thickness of the scleral tissue, increased quantities of glycosaminoglycans and collagen, the induction of the regularity of fibrils, and an increase in the diameter of fibrils in the sclera [19,20,21,22] (Figure 7).

The process of scleral remodeling is the final stage of the adaptation of the sclera to an eye with myopia. At this stage, the cytokine TGF-β plays an important role in regulating the changes in the extracellular matrix of the sclera associated with the development of myopia. A low TGF-β concentration results in decreased levels of collagen and proteoglycan synthesis. This causes the expression of α smooth muscle actin in fibroblasts, thereby causing the differentiation of fibroblasts into myofibroblasts [23].

### 4.3. The Effect of Citicoline on the MMP-2 Expression

Tissue remodeling is a complex process involving the synthesis and degradation of the extracellular matrix [24]. The changes in the structure of myopic sclera involve reduced levels of glycosaminoglycan and collagen and the irregular arrangement of collagen fibrils, thereby weakening the sclera’s biomechanical properties. Eyes with high or pathological myopia have thinner sclera with decreased posterior scleral thickness up to 31% compared to the normal adult sclera [25].

The balance of MMP-2 expression seems to have important roles in the metabolism of the extracellular matrix in the sclera. In 1996, Jones et al. observed a correlation between MMP-2 and myopia. The study found that the myofibrillar activity of gelatinase-A increased significantly for deprivation myopia [20,21]. In a study conducted by Zhao, it was found that, in visual deprivation conditions, there was an increase in the activation of free proenzyme MMP-2 along with a decrease in the level of tissue inhibitor of metalloproteinase (TIMP). Increased MMP-2 mRNA and decreased TIMP-2 mRNA levels in posterior sclera lead to ocular elongation caused by visual deprivation [21,26]. Due to its important role, a decreased level of MMP-2 may take part in the development of myopia, namely, the improvement of posterior scleral thickness, increased glycosaminoglycan and collagen levels, the induction of the regularity of collagen fibrils, an increased diameter of collagen fibrils, and the stabilization of scleral biomechanical properties [27].

This study also found that in the positive control group, the myopic rat models had the highest level of MMP-2 expression along with the longest axial length. Whereas, in the negative control and treatment group 3, the lowest level of MMP-2 expression and the most controlled axial length elongation were seen. From the statistical analyses, it was concluded that there was a weak and positive correlation between MMP-2 expression and axial length elongation. However, there is a possibility of the occurrence of axial length elongation due to the increasing level of MMP-2 expression.

The results are in accordance with a study by Mao showing that there was a significant increase in the retinal dopamine level in guinea pigs visually deprived for 10 days and injected with intraperitoneal citicoline (500 mg/kg BW; twice a day). These results demonstrated that an intraperitoneal injection of citicoline could retard the myopic shift induced by deprivation in guinea pigs, which was mediated by an increase in the retinal dopamine levels. Citicoline enhances the synthesis of phosphatidylcholine and stimulates the dopaminergic system in the retina. According to the study, the effect of the inhibition of citicoline on the progression of myopia is connected with an increase in the level of retinal dopamine [6].

Dopamine is the major neurotransmitter in the mammalian retina that acts through synapses and diffuses to farther targets including the choroid and sclera. The interaction of CDP-choline in the modulation of catecholaminergic transmission induces changes in the synthesis of retinoic acid in the choroid, whereas it affects the regulation of TGF-ß and extracellular matrix metabolism in the sclera, and MMP-2 is the dominant enzyme involved in the process [20,21].

Other neurochemical mechanisms are deprivation-upregulated, including the content and activity of retinal nitric oxide synthase in guinea pigs. In addition to its neuroprotective effect on dopaminergic neurons, citicoline could also reduce nitric oxide synthase levels and attenuate nitric oxide function in the neuronal damage of the retina and spinal cord. Thus, retinal nitric oxide may be involved in the citicoline inhibition of myopia’s progression [6].

### 4.4. The Effects of Citicoline on TGF-β1 Expression in the Rats’ Anterior and Posterior Scleral Tissues

The process of scleral remodeling is the final stage of adaptation of the sclera to myopia. At this stage, the cytokine TGF-β plays an important role in regulating changes in the extracellular matrix of the sclera associated with the development of myopia [23]. Transforming growth factor beta (TGF-β) is a multifunctional cytokine belonging to the superfamily of transformation growth factors that includes three different mammalian isoforms (TGF-β 1 to 3 and symbols HGNC TGFB1, TGFB2, and TGFB3) and many other signaling proteins. The TGF β protein is produced by all white blood cell lineages [28].

In 2007, research by Jobling et al. stated that the TGF-β isoform regulates scleral proteoglycan synthesis at various levels including that of the core protein, the initiation of the glycosaminoglycan chain (GAG), and sulfation. Based on the physiological levels of TGF-β in the sclera, TGF-β would likely mediate its proteoglycan effect through changes in the GAG side chain. Since TGF-β regulates scleral collagen and proteoglycan content, this cytokine is likely to be the main intra-scleral mediator for remodeling during the development of myopia [29].

In the treatment group administered with citicoline doses of 100 mg/kg BW and 200 mg/kg BW, there was a decrease in TGF-β1 expression that was not statistically significant compared to the positive control group. Administering a citicoline dose of 300 mg/kg BW yielded a significant increase in TGF-β1 expression compared to the positive control group. A statistically significant increase in TGF-β1 expression at a dose of 300 mg/kg BW compared to the positive control group indicated the potential of citicoline in controlling the progression of myopia in terms of increasing TGF-β1 expression in rats’ scleral tissue in both anterior and posterior sclera.

The results of the study that showed non-significant differences in the treatment groups at the doses of 100 mg/kg BW and 200 mg/kg BW could be due to the low dose of citicoline used, which was not significant with respect to promoting the increase in TGF-β1 expression. In addition, there is also another possible factor that may cause a decreasing effect, namely, the dual function effect of the TGF-β superfamily, including TGF-β1, which has an anti-proliferative and pro-proliferative role. From the results of this study, the secondary data could be obtained using linear regression, which showed that every 1 mg dose of citicoline can increase the expression of TGF-β1 by 0.030% in the anterior sclera and by 0.022% in the posterior sclera.

### 4.5. The Effects of Citicoline on Fibroblast Cell Proliferation through Ki-67 Expression in Rats’ Scleral Tissue 

The results of the study on Ki-67 expression to calculate the proliferation of fibroblast cells were obtained by immunohistochemical staining methods and calculations with the help of software with percentage units. The results showed that in the anterior scleral region, there was an increase in the expression of Ki-67, which indicated an increase in the degree of fibroblast proliferation in the positive control group followed by a decrease in the degree of fibroblast proliferation corresponding to the dose of citicoline given, which were statistically significant. Likewise, in the posterior pole region, there was a significant increase in fibroblast proliferation in the positive control group accompanied by a decrease in the level of fibroblast proliferation corresponding to the increasing dose of citicoline. 

The difference in the levels of fibroblast proliferation in the anterior sclera and posterior pole can be caused by a change, especially concerning biomechanics and biochemistry, which is more dominant in the posterior pole than in the anterior one due to the induction of myopia. Various studies previously mentioned, such as that conducted by Mc Brien et al. (2016), stated that there were changes in the scleral tissue of experimental animals after deprivation for 12 days, especially in the posterior pole. In other studies on the structural and biomechanical changes associated with myopia, it was stated that the posterior sclera experienced structural and biomechanical changes compared to the negative controls [11,18,27].

The results showed that the thickness of the scleral tissue was inversely proportional to the proliferation of fibroblasts after the myopic induction was carried out on the study rats. In the literature, it is stated that myopia occurs in the process of scleral remodeling, which affects the thickness of the scleral tissue and the proliferation of fibroblast cells of the scleral tissue. In the scleral tissue, there are cellular components and extracellular matrices in which there is a balance between the composition and the interactions between these components that determine the nature of the scleral tissue. Collagen is one component of the extracellular matrix, which makes up about 80% of the thickness of the sclera. Meanwhile, fibroblasts are cellular components that play an important role in fibroblast synthesis and express several proteins that influence the interactions between components of the scleral tissue. In studies regarding myopia, it was found that the amount of collagen decreased significantly in the myopic group compared to the control group. There are studies stating that the level of fibroblast proliferation decreased in the myopic group [25,28].

However, there are other studies that suggest an increase in fibroblast proliferation in myopia. This was stated by Hao et al. (2018), who showed that there is an increase in fibroblast proliferation in response to scleral tissue hypoxia that occurs in myopia. Simon et al. (2018) explained that the increase in fibroblast proliferation could be a response to tissue stress conditions due to the process of biochemical changes in the sclera. This notion explains the condition of the thickness of the sclera, which is inversely proportional to the proliferation of fibroblast cells [30,31].

The secondary outcome of this study illustrated that 1 mg of citicoline will increase the thickness of the sclera by 0.037 μm in the anterior sclera and 0.121 μm in the posterior pole. In addition, 1 mg of citicoline can reduce fibroblast proliferation by 0.003% in the anterior sclera and 0.004% in the posterior pole. It also shows that there is a different response of the scleral tissue between the anterior sclera and the posterior pole to the effect of the application of citicoline. In the literature, it is stated that the posterior pole is the thickest part of the sclera but has a more active mechanism and response in conditions of myopia. The literature suggests that there may be regional differences in growth hormone-specific expression associated with the response to myopia between the anterior and posterior sclera. Citicoline has a role in the scleral-remodeling process by increasing dopamine levels which results in signaling that effects the regulation of the extracellular components of the sclera (Figure 7). Furthermore, there will be a down-regulation of cytokines and specific growth hormones including FGFR-1, leading to a decrease in fibroblast proliferation. In addition, there is also the regulation of other influential cytokines such as TGF-ß to regulate the composition of the extracellular matrix. This is an essential condition for the balance regulation process to occur during an increase in extracellular matrix synthesis, including increased collagen and proteoglycan synthesis, increased levels of glycosaminoglycans (GAGs), and the decreased degradation of the extracellular matrix characterized by a decrease in the MMP and an increase in TIMP, which enables the thickness of the scleral tissue to increase [31,32].

From the research results, a dose of 200 mg/kg BW/day yielded the best results with respect to increasing scleral thickness and decreasing fibroblast proliferation to control the progression of myopia. This is in line with a study stating that an oral administration of 250 mg/kg BW causes a 59% increase in dopamine release through tyrosine hydroxylase stimulation. The release of this dopamine will provide a stop signal to the growth of the eyeball through the action of various mechanisms [20,22]. 

This is the first study observing the administration of oral citicoline in myopic rat models and evaluating their scleral tissues, specifically with regard to axial length elongation and MMP-2 expression. The limitation of this study is that refractive disorder was not measured using streak retinoscopy and other ocular parameters through biometry or ultrasonography due to the lack of available facilities. Consequently, this study only measured axial length after an enucleation procedure. Another limitation is that the observation of the diameter of the scleral collagen fibrils was not performed quantitatively. Since MMP-2 is secreted as a proenzyme that is in an inactive form, the immunohistochemistry method this study used has limitations because it only assessed the expression of MMP-2 secreted. The results of this study are expected to be a reference for the potential role of citicoline as a candidate agent in controlling the progression of myopia.

**Figure 7 biomedicines-10-02600-f007:**
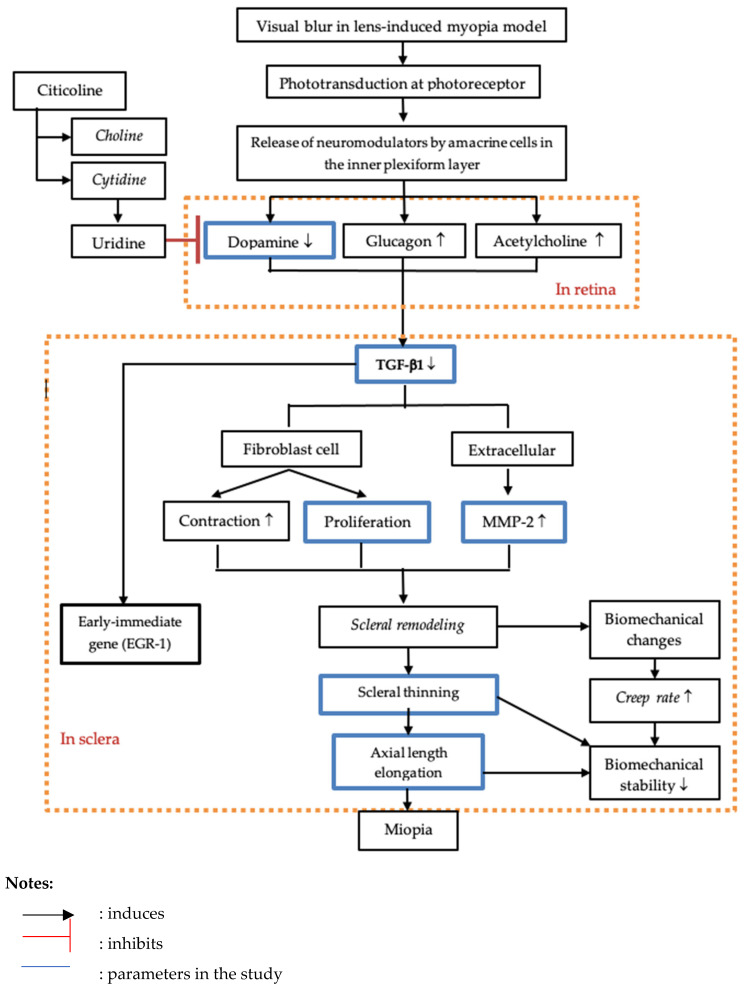
A schematic diagram of general pathway that may be important in the control of eye growth and how citicoline effects the scleral tissue. The first stage of growth signal cascades requires the linking of photoreceptors and dopaminergic amacrine cells inside the inner retina. Reduced stimulation of photoreceptors in LIM model decreases input to the dopaminergic amacrine cells [33]. The release of neurotransmitters by the retinal amacrine cells will convey a signal to the choroid and to the sclera [14,15,16]. The process of scleral remodeling is the final stage of adaptation of the sclera to an eye with myopia. The cytokine TGF-β plays an important role in regulating changes in the extracellular matrix of the sclera. A low TGF-β concentration results in decreased collagen and proteoglycan synthesis, thereby weakening the scleral extracellular matrix’s biomechanical properties. This causes the expression of α smooth muscle actin on fibroblasts, thus causing differentiation of fibroblasts into myofibroblasts [23]. The balance of MMP-2 expression seems to have important roles in the metabolism of extracellular matrix in the sclera [20,21]. Under LIM conditions, there was an increase in free proenzyme MMP-2 activated along with a decrease in tissue inhibitor of metalloproteinase levels (TIMP). Increased MMP-2 mRNA and decreased TIMP-2 mRNA levels in posterior sclera led to ocular elongation [21,26]. The administration of CDP-choline as a dopamine receptor agonist is expected to decrease fibroblast growth factor and inhibit MMP-2 activity such that there is a decrease in the number of scleral fibroblasts and a reduction in the degradation of the extracellular matrix. From these conditions, it is expected that there will be an improvement in the thickness of the scleral tissue, increased glycosaminoglycan and collagen levels, the induction of the regularity of fibrils, and an increase in the diameter of fibrils in the sclera [19,20,21,22].

## 5. Conclusions

Citicoline controls axial elongation and decreases MMP-2 and fibroblast proliferation significantly in the scleral tissue of myopic rat models. Citicoline also increases TGF-β1 expression and scleral tissue thickness significantly in myopic rat models. Therefore, citicoline has a potential role as a promising agent in controlling the progression of myopia. Further research is needed to determine the refractive disorder present after LIM using streak retinoscopy and biometry, to measure the activity of MMP-2, and to identify the exact pathways that mediate the effects.

## Figures and Tables

**Figure 1 biomedicines-10-02600-f001:**
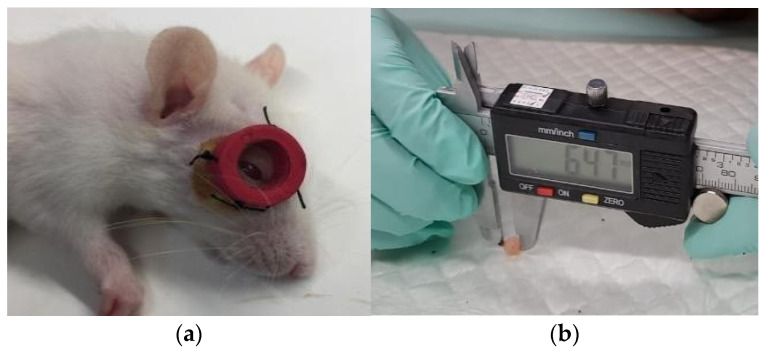
The procedure performed in this study. (**a**) Induction of myopia was conducted using the LIM method (attachment of S -10.00 D lens to the rat’s right eye); (**b**) axial length measurement used electronic digital calipers.

**Figure 2 biomedicines-10-02600-f002:**
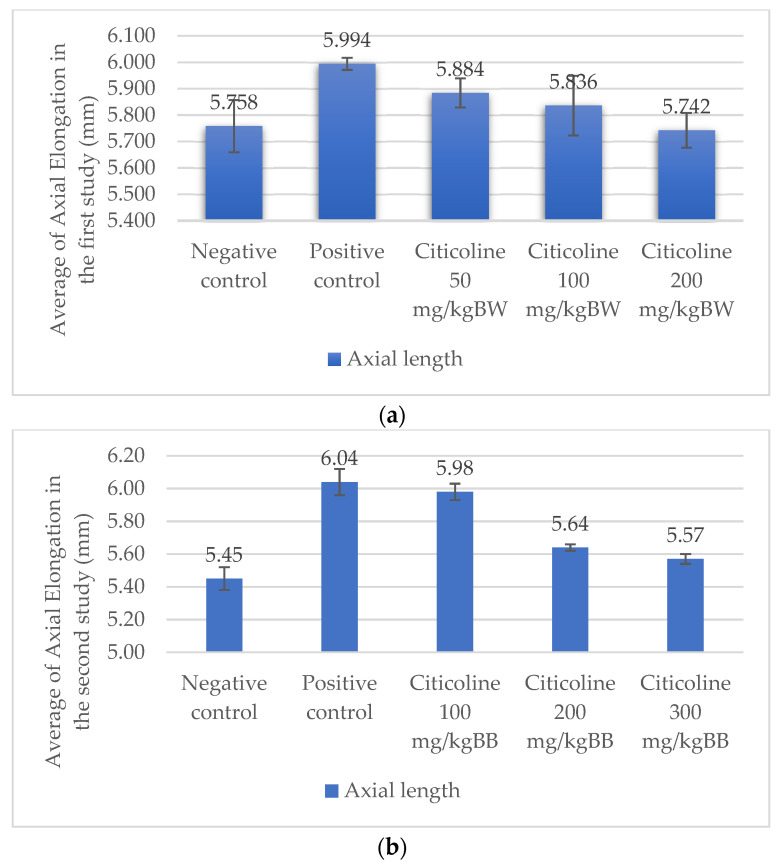
(**a**) A chart displaying the axial elongations in the first study. Based on Tukey test, the negative control (5.758 ± 0.099 vs. 5.994 ± 0.023, *p* = 0.001) and treatment group administered with the dose of 100 mg/kg BW (5.836 ± 0.113 vs. 5.994 ± 0.023 vs, *p* = 0.032) and 200 mg/kg BW (5.742 ± 0.066 vs. 5.994 ± 0.023, *p* = 0.000) show the most controlled axial elongation compared to positive control group. (**b**) A chart of the axial elongations in the second study. Based on Tukey test, the negative control (5.45 ± 0.07 vs. 6.04 ± 0.08, *p* = 0.000) and the treatment groups with a dose of 200 mg/kg BW (5.64 ± 0.02 vs. 6.04 ± 0.08, *p* = 0.000) and a dose of 300 mg/BW (5.57 ± 0.03 vs. 6.04 ± 0.08, *p* = 0.000) show the most controlled axial elongation compared to positive control group. In both studies, the positive control group has the longest axial length compared to the other groups.

**Figure 3 biomedicines-10-02600-f003:**
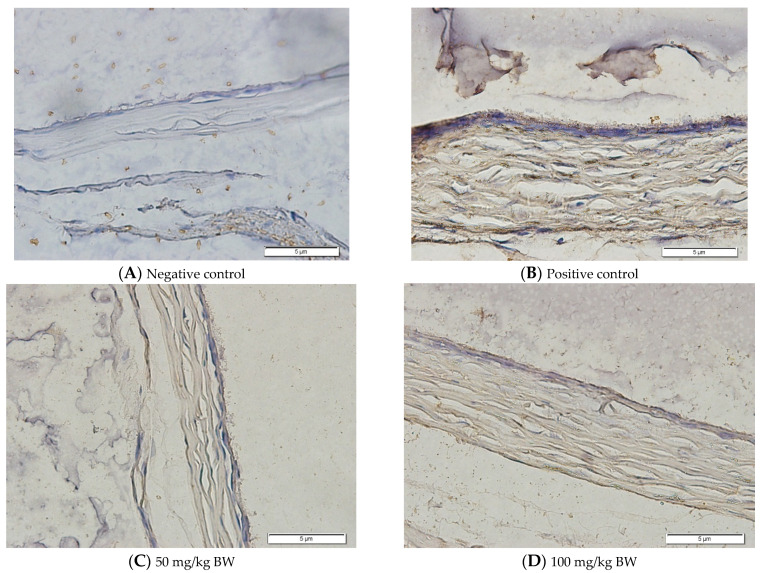
Rats’ Scleral Tissue after Antibody Anti-MMP-2 Staining (Left) (**A**–**E**). (**A**) Negative control group; (**B**) Positive control group; (**C**) Treatment groups with a citicoline dosage of 50 mg/kg BW; (**D**) 100 mg/kg BW; (**E**) 200 mg/kg BW; (**F**) The chart of mean MMP-2 expression in the scleral tissue. Based on Tukey test, the positive control group has the highest MMP-2 expression compared to the negative control group (70.88 ± 18.28 vs. 15.87 ± 6.17, *p* = 0.000), and treatment groups with a dose of 100 mg/kg BW (70.88 ± 18.28 vs. 31.64 ± 23.28, *p* = 0.004) and 200 mg/kg BW (70.88 ± 18.28 vs. 18.38 ± 7.75, *p* = 0.000).

**Figure 4 biomedicines-10-02600-f004:**
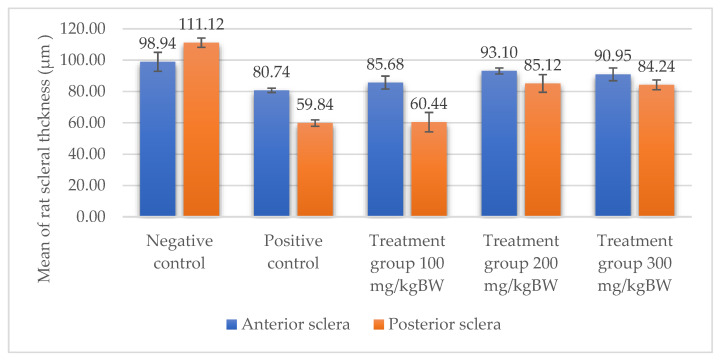
Mean thickness of the rats’ scleral tissue. From Tukey test, we found that the negative control group shows greater thickness in the anterior sclera and posterior pole compared to the positive control group (98.94 ± 6.06 μm vs. 80.74 ± 1.39 μm, *p* = 0.002 in the anterior sclera and 111.12 ± 2.99 μm vs. 59.84 ± 2.02 μm, *p* = 0.000 in the posterior pole). The treatment groups administered with 200 mg/kg BW/day (93.10 ± 1.93 μm vs. 80.74 ± 1.39 μm, *p* = 0.002 in the anterior sclera and 85.12 ± 5.62 μm vs. 59.84 ± 2.02 μm, *p* = 0.000 in the posterior pole) and 300 mg/kg BW/day (90.95 ± 4.07 μm vs. 80.74 ± 1.39 μm, *p* = 0.000 in the anterior sclera and 84.24 ± 3.15 μm vs. 59.84 ± 2.02 μm, *p* = 0.000 in the posterior pole) show significant differences compared to the positive control group.

**Figure 5 biomedicines-10-02600-f005:**
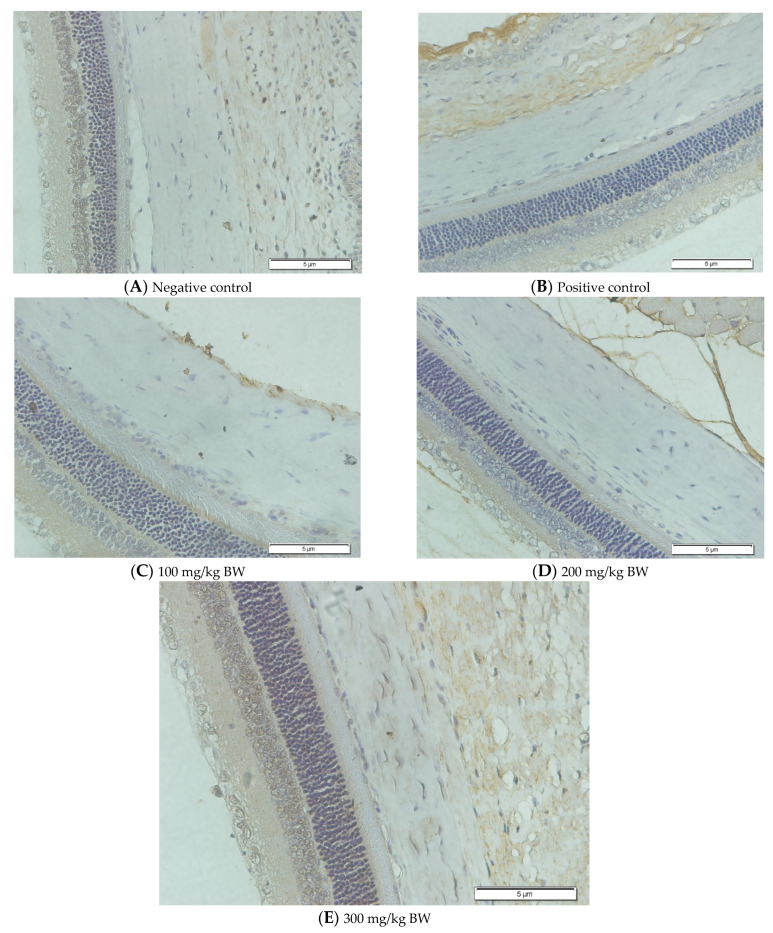
The Rats’ Scleral Tissue after Antibody Anti-TGF-β1 Staining (Upper) (**A**–**E**). (**A**) The negative control group; (**B**) the positive control group; (**C**) the treatment groups administered with a citicoline dosage of 100 mg/kg BW; (**D**) 200 mg/kg BW; (**E**) 300 mg/kg BW. (**F**) A chart illustrating the mean TGF-β1 expressions in the anterior and posterior scleral tissue (Lower). From Tukey test, we found that the treatment group administered with 300 mg/BW/day has the highest TGF-β1 expression compared to the negative control group (12.86 ± 2.43 vs. 4.55 ± 0.77, *p* = 0.000 in the anterior sclera and 10.71 ± 4.56 vs. 5.01 ± 2.94, *p* = 0.042 in the posterior sclera), the positive control group (12.86 ± 2.43 vs. 3.34 ± 0.54, *p* = 0.000 in the anterior sclera and 10.71 ± 4.56 vs. 3.40 ± 1.75, *p* = 0.006 in the posterior sclera), the treatment group administered with 100 mg/BW/day (12.86 ± 2.43 vs. 2.26 ± 0.80, *p* = 0.000 in the anterior sclera and 10.71 ± 4.56 vs. 3.12 ± 1.20, *p* = 0.005 in the posterior sclera), and the treatment group administered 200 mg/BW/day (12.86 ± 2.43 vs. 3.77 ± 3.15, *p* = 0.000 in the anterior sclera and 10.71 ± 4.56 vs. 3.30 ± 3.00, *p* = 0.006 in the posterior sclera).

**Figure 6 biomedicines-10-02600-f006:**
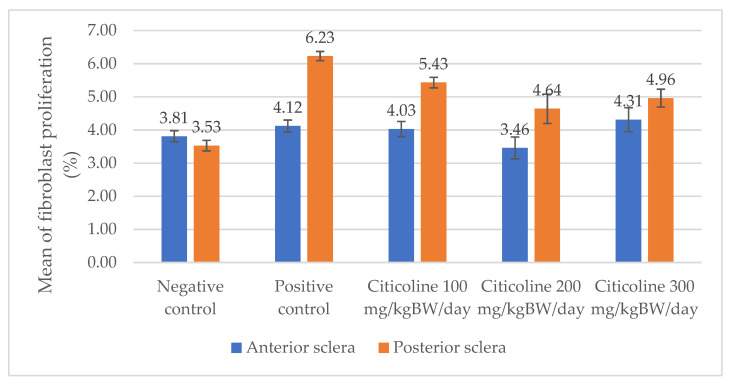
Mean fibroblast proliferation of the rats’ scleral tissue. Based on Tukey test, the positive control group shows a higher level of fibroblast proliferation compared to the negative control group (4.12 ± 0.18 vs. 3.81 ± 0.17, *p* = 0.227, in the anterior sclera and 6.23 ± 0.14 vs. 3.53 ± 0.16, *p* = 0.000, in the posterior sclera). Among the treatment groups, the treatment group administered with 200 mg/kg BW/day shows the lowest level of fibroblast proliferation compared to the positive control group (3.46 ± 0.33 vs. 4.12 ± 0.18, *p* = 0.001 in the anterior sclera and 4.64 ± 0.44 vs. 6.23 ± 0.14, *p* = 0.000 in the posterior sclera).

**Table 1 biomedicines-10-02600-t001:** Measurements of the rats’ scleral tissue thicknesses using HE-staining method.

Regio	Negative Control	Positive Control	Citicoline 100 mg/kg BW/day	Citicoline 200 mg/kg BW/day	Citicoline 300 mg/kg BW/day
Anterior sclera	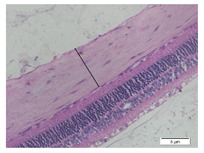	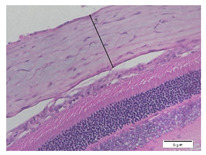	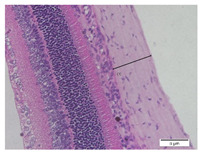	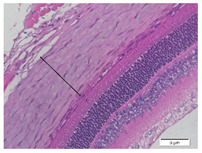	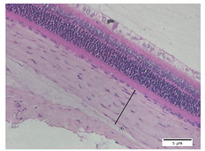
	(a)	(b)	(c)	(d)	(e)
Posterior pole	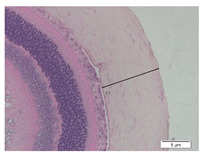	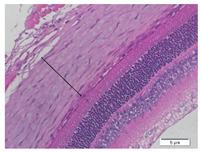	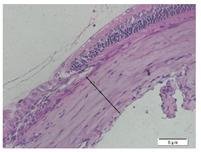	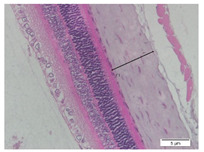	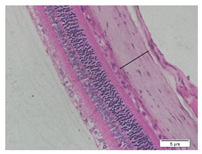
	(f)	(g)	(h)	(i)	(j)

**Table 2 biomedicines-10-02600-t002:** Measurements of fibroblast proliferation using Ki-67.

Regio	Negative Control	Positive Control	Citicoline 100 mg/kg BW/day	Citicoline 200 mg/kg BW/day	Citicoline 300 mg/kg BW/day
Anterior sclera	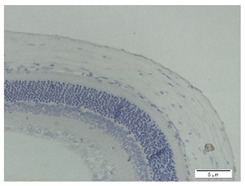	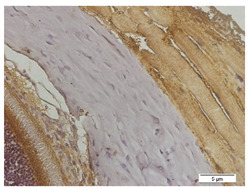	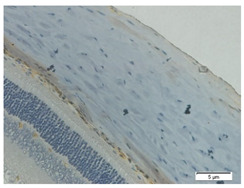	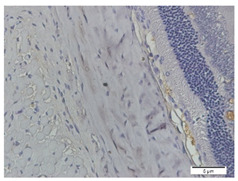	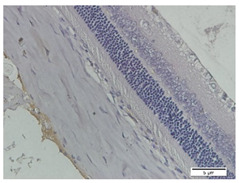
	(a)	(b)	(c)	(d)	(e)
Posterior pole	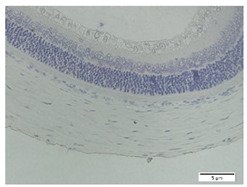	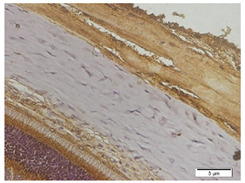	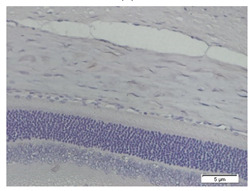	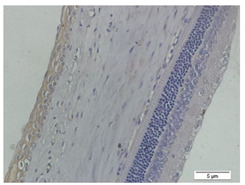	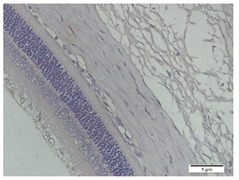
	(f)	(g)	(h)	(i)	(j)

## Data Availability

Data is contained within the article.

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
