# Peer review of "The Effect of Citicoline on the Expression of Matrix Metalloproteinase-2 (MMP-2), Transforming Growth Factor-β1 (TGF-β1), and Ki-67, and on the Thickness of Scleral Tissue of Rat Myopia Model"

_biomedicines, 2022, doi:10.3390/biomedicines10102600_

Round 1

Reviewer 1 Report

In their manuscript entitled "The Effect of Citicoline on the Expression of Matrix Metalloproteinase-2 (MMP-2), Transforming Growth Factor-ß1 (TGF- ß1), and Ki-67, and on the Thickness of Scleral Tissue of Rat Myopia Model"

the authors describe an very interesting in vivo study.

In gernerell, the study design is very well and the manuscript contains all important informations.

However, for quality aspects and meet the generel rules for high quality publication, I recommend the following revision:

- the figure legend should also state (even the authors stated that they used the ANOVA test, T test ), it should be stated again in the figure legends (als the p value - even if it is written in the text) so that a reade can directly "see" the significance.

- The conclusion is too short, please broaden it and mention future investigations - and what research is still nessescary

- cleary fornulate the hypotheses H0 and H1

Overall, I recommend for publication after these minor revision

Author Response

Response to Reviewer 1 Comments

In their manuscript entitled "The Effect of Citicoline on the Expression of Matrix Metalloproteinase2 (MMP-2), Transforming Growth Factor-ß1 (TGF- ß1), and Ki-67, and on the Thickness of Scleral

Tissue of Rat Myopia Model" the authors describe an very interesting in vivo study.

In gernerell, the study design is very well and the manuscript contains all important informations.

However, for quality aspects and meet the generel rules for high quality publication, I recommend the following revision:

Point 1: the figure legend should also state (even the authors stated that they used the ANOVA test, T test ), it should be stated again in the figure legends (als the p value - even if it is written in the text) so that a reade can directly "see" the significance.

Response 1: Thank you for the advise. We already revised all the figure legends as suggested by reviewers.

Point 2: The conclusion is too short, please broaden it and mention future investigations - and what research is still nessescary.

Response 2: 

Further research is already mentioned in the end of discussion (4.5). We agree with the reviewer and move it into conclusion. It was previously stated in line 585 (now line 748-750 ) “Further research is needed to determine the refractive disorder after LIM using streak retinoscopy and biometry to measure the activity of MMP-2 and to identify the exact pathways that mediate the effects.” Conclusion: 

Citicoline controls axial elongation and decreases MMP-2 and fibroblast proliferation significantly in the scleral tissue of myopic rat models. Citicoline also increases TGF-b1 expression and scleral tissue thickness significantly in myopic rat models. Therefore, citicoline has a potential role as a promising agent in controlling the progressivity of myopia. Further research is needed to determine the refractive disorder after LIM using streak retinoscopy and biometry to measure the activity of MMP2 and to identify the exact pathways that mediate the effects. Further research is needed to determine the refractive disorder after LIM using streak retinoscopy and biometry to measure the activity of MMP-2 and to identify the exact pathways that mediate the effects.

Point 3: cleary fornulate the hypotheses H0 and H1

Response 3: We agree with the reviewer and have added this statement in the end of the introduction (line 81-83): “We propose that citicoline reduces MMP-2 expression and fibroblast proliferation and increase TGF-b1 expression and scleral tissue thickness and, therefore, may be useful for inhibiting the progression of myopia.” 

Overall, I recommend for publication after these minor revision

Reviewer 2 Report

The authors have done a good work on providing evidence that Citicoline is involved in controlling myopia by studying the expression of MMP-2, TGFb1, and Ki-67, and the thickness of scleral tissue of rat myopia model. However, the authors need to specify the significance in all the figures. It is mentioned in the text but not shown in the figures. In addition, a graphical abstract showing how Citicoline acts to increase the scleral tissue thickness and what pathways might be involved would benefit this study. 

Author Response

Response to Reviewer 2 Comments

The authors have done a good work on providing evidence that Citicoline is involved in controlling myopia by studying the expression of MMP-2, TGFb1, and Ki-67, and the thickness of scleral tissue of rat myopia model. 

Point 1: However, the authors need to specify the significance in all the figures. It is mentioned in the text but not shown in the figures. 

Response 1: Thank you for the reviewer’s advice. We already revised all the figure legends and mentioned the significance in the figures as suggested by reviewers.

Point 2: In addition, a graphical abstract showing how Citicoline acts to increase the scleral tissue thickness and what pathways might be involved would benefit this study. 

Response 2: 

We agree with the reviewer and have added a scheme about how citicoline may act to cause the effects on scleral tissue. We add the scheme in discussion as figure 7 in the end of the discussion. 
